# Qualitative and Quantitative Determination of Decoquinate in Chicken Tissues by Gas Chromatography Tandem Mass Spectrometry

**DOI:** 10.3390/molecules28093875

**Published:** 2023-05-04

**Authors:** Shuyu Liu, Yayun Tang, Yang Lu, Yawen Guo, Kaizhou Xie, Fanxun Guan, Pengfei Gao, Yali Zhu, Yuhao Dong, Tao Zhang, Genxi Zhang, Guojun Dai, Xing Xie

**Affiliations:** 1College of Animal Science and Technology, Yangzhou University, Yangzhou 225009, China; mz120211479@stu.yzu.edu.cn (S.L.); tyy199727@163.com (Y.T.); mz120191020@yzu.edu.cn (Y.L.); dx120200135@yzu.edu.cn (Y.G.); daolao71935@163.com (F.G.); mz120201383@yzu.edu.cn (P.G.); mz120211463@stu.yzu.edu.cn (Y.Z.); zhangt@yzu.edu.cn (T.Z.); gxzhang@yzu.edu.cn (G.Z.); daigj@yzu.edu.cn (G.D.); 2Joint International Research Laboratory of Agriculture & Agri-Product Safety, Yangzhou University, Yangzhou 225009, China; 3College of Veterinary Medicine, Nanjing Agricultural University, Nanjing 210095, China; dongyuhao@njau.edu.cn; 4Key Laboratory of Veterinary Biological Engineering and Technology, Institute of Veterinary Medicine, Jiangsu Academy of Agricultural Sciences, Ministry of Agriculture, Nanjing 210014, China

**Keywords:** decoquinate, gc-ms/ms, derivatization, chicken tissues, acetic anhydride

## Abstract

A novel precolumn derivatization–GC-MS/MS method was developed for the determination of decoquinate residues in chicken tissues (muscle, liver, and kidney). The samples were extracted and purified by liquid–liquid extraction combined with solid-phase extraction and derivatized with acetic anhydride and pyridine. The recovery rates for decoquinate were 77.38~89.65%, and the intra-day and inter-day RSDs were 1.63~5.74% and 2.27~8.06%, respectively. The technique parameters meet the necessities for veterinary drug residue detection in China, the US, and the EU. Finally, the method was applied to analyze tissues of 60 chickens bought from a neighborhood supermarket, and solely one sample of chicken muscle contained 15.6 μg/kg decoquinate residue.

## 1. Introduction

Fowl coccidiosis is an infectious disease caused by one or more species of *Eimeria* spp. parasitic in the epithelial cells of the chicken intestine, mainly caused by the tender *Eimeria* spp. parasitic in the cecum of chicks and the toxic *Eimeria* spp. parasitic in the small intestine [1]. Decoquinate (DQ) is a quinoline-based chemical anticoccidial drug commonly used for the prevention and long-term treatment of coccidiosis in poultry [2]. DQ acts on the cytochrome bc1 complex of *Eimeria* at the early stage of its life history, blocks electron transfer in the cytochrome system in the mitochondria, inhibits the growth of ascospores and first-generation lysosomes, prevents the further development of *Eimeria* from causing more damage to the intestinal tract of animals, and thus achieves the effect of prevention and treatment of coccidiosis [3]. Studies have shown that DQ is widely used for the control of coccidiosis in animals such as pigs, cattle, sheep, and rabbits because of its rapid metabolism, low toxicity, and good tolerability [4,5,6,7,8,9,10,11]. However, the nonstandardized use and abuse of drugs will cause veterinary drug residues in animal foods and damage human health. As a result, maximum residue limits (MRLs) for DQ have been established by different countries and organizations (Table 1) [12,13,14,15].

At present, the detection techniques accessible for the residues of DQ in animal food mainly include high-performance liquid chromatography–ultraviolet detection (HPLC-UV) [16], high-performance liquid chromatography–diode array detection (HPLC-DAD) [17], high-performance liquid chromatography fluorescence detection (HPLC-FLD) [18,19] and liquid chromatography–mass spectrometry (LC-MS) [20,21,22,23,24]. However, gas chromatography–tandem mass spectrometry (GC-MS/MS) methods for the determination of DQ residues in foods of animal origin have not been reported.

Combined GC and MS methods utilize the excellent separation ability of GC and the identification capability of MS [25]. The technique has the advantages of high resolution, high sensitivity, good accuracy, and lower cost than LC-MS. Compared with GC-MS, GC-MS/MS generates secondary cleavage of the compounds, better separation, and increased analysis speeds. The technique has wide application prospects in food safety, environmental monitoring, and drug analyses.

In this study, we mainly referred to the Chinese limits for DQ in chicken tissues. A combination of liquid–liquid extraction (LLE) and solid-phase extraction (SPE) was used for the extraction and purification of the samples. Acetic anhydride was used as a derivatization reagent in acetylation reactions of DQ in the presence of pyridine. Detection was performed with an optimized GC/MS/MS method. The aim of this study was to establish a pre-column derivatization-GC/MS/MS technique for the detection of DQ residues in chicken tissues (muscle, liver, and kidney), and to provide technical support for the detection of DQ residues in foods of animal origin.

## 2. Results and Discussion

### 2.1. Confirmation of Derivative Products

In this experiment, the reaction mechanism of DQ and 1,8-diazabicycloundec-7-ene (DBU) was referred to [26], and acetic anhydride, which is commonly used in the laboratory and has good stability, was used as a derivative reagent instead of DBU. DQ with a boiling point up to 517.9 °C was derived into acetylated DQ, which can be directly analyzed and detected using GC-MS/MS.

For this study, 1.0 mL of standard solution with a concentration of 10.0 μg/mL DQ was placed in a centrifuge tube, 200 μL of pyridine and 100 μL of acetic anhydride were added sequentially, and the reaction was carried out at room temperature, and protected from light for 4 h. At the end of the reaction, instrumental analysis was performed. After the derivatized DQ standard solution entered the GC-MS/MS system, ion source bombardment (EI), full scan, and selective ion scan (SIM) modes were performed to collect data, and the full scan chromatogram and mass spectrum of acetylated DQ were obtained (Figure 1). According to the chemical structure of the derivative, the mass spectrum corresponding to each chromatographic peak was analyzed separately, and the derivative was preliminarily confirmed to be acetylated DQ.

After the retention times of the derivatized products were preliminarily determined, a concentration gradient test of the drug and the derivatized reagent was performed. DQ standard solutions were prepared at a concentration of 1.0 mg/mL and diluted to 500.0 μg/mL, 250.0 μg/mL, 125.0 μg/mL, 62.5 μg/mL, and 31.25 μg/mL standard working solutions. A total of 1.0 mL of standard working solutions at six concentrations were taken, and six replicate solutions were prepared for each concentration. Subsequently, 200 μL of pyridine and 100 μL of acetic anhydride were added sequentially. After the derivatization reaction was completed, the samples were analyzed by instrument. A standard curve was established with DQ concentration as the abscissa (x) and the peak area of the target peak as the ordinate (y) (Figure 2). The linear relationship was excellent, with a determination coefficient (R^2^) of 0.9996. This further confirmed that the derivative was acetylated DQ.

### 2.2. Determination of Parent and Product Ions

The full-scan mass spectrum of the derivative was observed, and the parent ion selected was the *m*/*z* 231.1 ion with the largest mass charge ratio and the highest abundance. After the determination of the parent ion, Auto SAM was used to optimize the instrument method to screen out the optimal collision energy. Information on the intensity of daughter ion fragments observed after the parent ion was produced by different collision energies (Figure 3). Two fragment ions of *m*/*z* 230.1 and *m*/*z* 229.1 showed the optimal collisional energy and abundance ratio. These fragments were stable, not easily interfered with by other substances, and screened as daughter ions, the parent ion and that daughter ion form a monitoring ion pair with each other (Table 2).

### 2.3. Optimization of Derivative Conditions

#### 2.3.1. Optimum Dosage of Acetic Anhydride

To ensure the accuracy of quantitative results, the optimal reaction conditions for DQ and acetic anhydride were optimized and finally determined. The amount of acetic anhydride was plotted on the horizontal axis, and the peak area of the derivative was plotted on the vertical axis (Figure 4). The peak area of the derivative product showed an increasing trend with amounts of acetic anhydride between 50–150 μL, and the maximum peak area of the derivative was reached when the additional amount of acetic anhydride was 150 μL. The peak area was unstable and showed a downward trend with amounts of acetic anhydride between 150–250 μL. The peak area then exhibited a slowly rising trend with the amount of acetic anhydride between 250 μL and 350 μL, but these peak areas were always smaller than that at the 150 μL addition of acetic anhydride. As the amount of acetic anhydride was increased to 400 μL, the peak area of the derivative product gradually decreased. As a result, the optimal dose for the reaction of acetic anhydride with DQ was 150 μL.

#### 2.3.2. Optimum Pyridine Dosage

The graph was plotted with the amount of pyridine as the horizontal coordinate and the peak area of the derivative as the vertical coordinate (Figure 5). As shown in Figure 5, when the pyridine dosage was 300 μL, the peak area of the derivative reached its maximum. With the increase in the amount of pyridine, there was no significant change in the peak area of the derivatives. Therefore, the optimum dosage of pyridine for our experiment was 300 μL.

#### 2.3.3. Optimal Derivation Time

After determining the amount of acetic anhydride and pyridine, the reaction time of the derivatization was further optimized. Curve with derivation time as the horizontal coordinate and peak area of the quantitative ion pair of the derivative as the vertical axis. As shown in Figure 6, the peak area of the derivative slowly increased with time when the derivation time was in the range of 0.5–3.5 h and reached a maximum of 3.5 h. Subsequently, the peak area of the derivative product gradually stabilized and did not change significantly after 3.5 h.

As shown in Figure 4, Figure 5 and Figure 6, the optimal derivation conditions for DQ and acetic anhydride were 150 μL acetic anhydride in the dark for 3.5 h with 300 μL pyridine.

### 2.4. Sample Pretreatment Optimization

#### 2.4.1. Selection of Extraction Reagents

In this study, the extraction effects of acetonitrile [24,27], acetonitrile:ethyl acetate (1:1, *V*/*V*), and 4% acetic acid–acetonitrile solution on DQ in chicken, liver, and kidney were compared. Acetonitrile had the optimal extraction effect on DQ, and the recoveries from the three matrices were 85.29%~87.03% (Table 3). After the extraction reagents were determined, the effect of extraction reagents at multiple concentrations on the recovery was investigated. The results of five comparative experiments showed that (Figure 7) 100% acetonitrile was the optimal extraction reagent in this experiment.

#### 2.4.2. Selection of Extraction Times

This experiment investigated the effect of the number of extractions on the experimental results. The three matrices of chicken tissues (muscle, liver, and kidney) were extracted once, twice, and three times, respectively, to observe the changes in the peak area of acetylated DQ. The DQ peak area was maximized when the samples were extracted twice in the three matrices, and the peak area did not change significantly when the samples were extracted three times. However, the peak area occasionally decreased due to an increase in the number of extractions. Therefore, the optimal number of extractions in our experiment was two (Figure 8).

#### 2.4.3. Selection of Solid Phase Extraction Column

In this experiment, the chemistry of DQ and the method of use reported in the literature to date were the primary factors in the selection of the solid phase extraction column. Chen et al. compared three SPE columns, HLB, silica gel column, and MAX, for the cleanup of DQ in chicken meat, and the experimental results showed that the HLB column was the best [28]. Kim et al. selected HLB columns for the cleanup of two anticoccidials in beef and chicken muscle, and the recoveries of DQ ranged from 78.5% to 107.1% [29]. Beatriz et al. analyzed the residues of DQ in milk by HPLC-FLD detection and selected Strata^TM^-X Polymeric Sorbent Cartridges to purify the sample, with an average recovery rate of 88.7% [30]. Cleanert PEP has a strong adsorption effect on various polar compounds and wide applicability, which is similar to that of the Waters Oasis HLB column, but the price is 3.7 times lower than that of the HLB column.

In this experiment, the purification and enrichment effects of Waters Oasis HLB (60 mg, 3 mL), Strata^TM^-X (60 mg, 3 mL), and Cleant PEP (60 mg, 3 mL) SPE columns were compared. The comparison results are presented in Figure 9, where recovery of acetylated DQ was highest when samples were purified using HLB columns.

### 2.5. GC-MS/MS Analysis Optimization

The derivatized decoquinate produces a new compound, so the temperature programming cannot be set according to the boiling point of the original drug, and there is no available literature for reference. The collection time at each temperature can only be extended to the maximum extent during the preliminary test to extrapolate the approximate range of the boiling point from the retention time of the derivative product. The test was initiated by programming the initial temperature to 100 °C for 1 min, ramped at 30 °C/min to 220 °C for 1 min, and ramped at 20 °C/min to 280 °C for 16 min. At the time of data collection, the retention time of the target peak was approximately 23.10 min, which provided a derivative boiling point of approximately 280 °C. Next, by changing the temperature programming to shorten the target peak time, the analysis efficiency could be improved. The program was adjusted to initially maintain 100 °C for 1 min, ramped at 15 °C/min to 200 °C for 2 min, and ramped at 20 °C/min to 280 °C for 7 min. With this adjustment, the chromatographic peak of the derivative product always appeared during the second data acquisition. Therefore, the temperature programming chosen for our test is shown in Table 4.

The MS conditions mainly included ionization mode, ion source and transmission line temperatures, solvent delay time, and acquisition data mode. Temperatures that are too high or too low for the transmission line and the ion source can directly affect the analysis results of the sample and reduce the service life of the ion source. The scanning range and acquisition time were mainly selected based on the chemical properties of the target and using a comparison from the results of multiple experiments. In this experiment, the qualification was mainly performed using the molecular weight of the derivative products combined with the mass-to-charge ratio in the mass spectrum and the concentration gradient test results; the mass spectrum conditions were adjusted after the derivative products were determined. The full scan range was *m*/*z* 50–550, and the analyte was identified, followed by quantitative scanning using Auto SRM. The final mass spectrometric conditions from the qualitative and quantitative scan results are presented in Section 2.2.

### 2.6. Method Validation

#### 2.6.1. Linearity

As shown in Figure 10, Figure 11 and Figure 12, the linear regression equations for the DQ of chicken muscle, liver, and kidney were y = 564.28x + 37,020, y = 530.15x + 30,524, and y = 559.75x + 19,541, respectively. Their correlation coefficients (R^2^) were all greater than 0.9990.

#### 2.6.2. LOD and LOQ

The concentrations at which the signal-to-noise ratios (S/N) of the target compound were 3 and 10 were used as the LOD and LOQ for the target compound in the chicken tissue (muscle, liver, and kidney) samples, respectively. Table 5 shows that the LOD and LOQ of DQ in chicken muscle were 2.2 µg/kg and 4.9 g/kg, respectively. The LOD and LOQ of DQ in chicken liver were 4.3 µg/kg and 8.2 g/kg, respectively. The LOD and LOQ of DQ in chicken kidneys were 3.7 µg/kg and 6.3 µg/kg, respectively. Therefore, the method has high sensitivity and accuracy.

#### 2.6.3. Matrix Effect

The matrix effects for DQ in chicken tissues were between −19.64% and −15.58% when using GC-MS/MS detection. As shown in Table 6, all of the targets showed weak ion suppression effects, which may have arisen because some of the water-soluble proteins were not effectively removed and competed with the targets for H^+^ during the ionization process. In this experiment, the samples were subjected to repeated extractions with ultrasonication, shaking, and centrifugation, which ensured that all of the analytes were as fully dissolved in the extractant as possible; most of the matrix was separated out by centrifugation, which greatly improved the accuracy of the test results.

#### 2.6.4. CCα and CCβ

The results are presented in Table 4, and CCα and CCβ for the three matrices were calculated from the equation in Section 3.6.4. When the detected concentration of chicken samples was more than 1032.0 μg/kg, the probability of noncompliance of the positive sample was 95%; when the detected concentration of samples was more than or equal to 1034.1 μg/kg, a 95% probability that a sample of 1000 μg/kg will be detected. When the concentration of the detected chicken liver sample was more than 2042.0 μg/kg, the probability of noncompliance of the positive sample was 95%; when the detected concentration of samples was more than or equal to 2049.0 μg/kg, there is a 95% probability that a sample of 2000 μg/kg will be detected. When the detected concentration of the chicken kidney sample was higher than 2047.0 μg/kg, the probability of noncompliance of the positive sample was 95%. When the detected concentration of the sample was greater than or equal to 2054.1 μg/kg, there was a 95% probability that a sample of 2000 μg/kg will be detected.

#### 2.6.5. Recovery and Precision

Four concentrations of DQ standard working solutions (LOQ, 0.5 MRL, 1.0 MRL, and 2.0 MRL) were added to the blank chicken muscle, liver, and kidney. A test for adding the standard substance to the blank sample was conducted using our sample pretreatment and derivation methods. Table 7 shows the relevant data on the addition recovery rate and precision of DQ.

When the concentration range of the DQ standard added to the blank chicken muscle was 4.9~2000.0 μg/kg, the spiked recoveries for DQ at the four concentrations in chicken were within the range of 77.38~88.33%, the intraday relative standard deviations (RSDs) were 2.14~5.09%, and the interday RSDs were 2.27~6.94%. When the concentration range of DQ in the blank chicken liver was 8.2~4000 μg/kg, the spiked recovery rate was 78.33~89.65%, the intraday RSD was 2.26~5.56%, and the interday RSD was 3.47~6.62%. The recoveries of DQ were 79.15~89.10% when the concentration range of the standard substance was 6.3~4000 μg/kg in the kidney of blank chickens. The intraday RSD was 1.63~5.74%, and the interday RSD was 2.47~8.06%. As shown in Figure 13, the peak emergence time for the acetylated DQ was approximately 17.40 min, and the peak shape was sharp with no tailing. In addition, it was well separated from the impurity peak and could be used for quantitative analysis.

### 2.7. Stability of Standard Solutions and Derivatives

As shown in Figure 14, the standard solution of DQ can be stored stably at −70 °C for 56 days; decomposition of the standard solution tends to occur after the 56th day.

The stability of the derivatives was examined after determining the retention time of the derivatives. In this test, the stability of acetylated DQ was investigated in three storage environments: at room temperature, in a 4 °C refrigerator, and in a −20 °C refrigerator. Figure 15 shows that the stability of acetylated DQ is best within 48 h when it is stored in a sealed −20 °C refrigerator. Throughout the experiment, all samples were tested on the instrument immediately after the derivatization reaction. However, in special cases, the derivatization product could be redissolved in chloroform after nitrogen blowing and stored for 48 h in a refrigerator at −20 °C.

### 2.8. Comparison with Other Methods

The main methods used for detecting DQ residues in animal foods are currently instrumental assays, including HPLC-UV [31], HPLC-MS [32], HPLC-MS/MS [33], UHPLC-MS/MS [34], LC-MS/MS [35], and HPLC-FLD [8]. As seen in Table 8, the mass spectrometric method exhibited the highest sensitivity, but GC-MS/MS is less expensive than HPLC-MS/MS and uses an inert gas as the mobile phase, which eliminates the need for frequent preparation of liquid mobile phases. The LC-MS/MS method established by Li et al. for simultaneous detection of 19 anticoccidial residues in eggs requires two solid phase extraction columns, a Waters Sep PaV Alumina N Cartridge and a Waters Oasis HLB, in order to clean up the samples after extraction in this method, which is costly [36]. The time required for sample pretreatment and the quality of the method are also important factors that affect the recovery rate. Lin et al. developed a UPLC-MS/MS for the quantitative analysis of residual DQ in bovine liver. Repeated extraction and centrifugation were needed in the pretreatment and the margin of the extracting solution needed to be strictly controlled during concentration. More manual steps correspond to a greater impact on the test results [34]. Matus et al. established an HPLC-MS/MS method for the detection of residues of 17 anticoccidial drugs in chicken to fully use the high sensitivity of mass spectrometry, but the sample pretreatment method was complicated and time-consuming and required high-speed shaking on a horizontal shaker for 30~45 min [33]. The GC-MS/MS detection method established in this study to determine the DQ residues in chicken tissues is more sensitive than the reported HPLC-UV instrumental analysis method [17,33], and the operations are simple. Therefore, we have developed a GC-MS/MS method for the determination of DQ in chicken tissues, which meets the needs of different laboratories.

### 2.9. Real Sample Analysis

The method developed in this study was used to analyze 60 chicken tissue samples purchased from local supermarkets. After the samples were processed and analyzed using GC-MS/MS, the results showed that the target compound was not found in the chicken livers and kidneys of these samples, and only one chicken muscle sample contained 15.6 μg/kg DQ. Thus, the GC-MS/MS method can be used for the quantitative detection of DQ residues in chicken tissues, which provides a new method for the determination of DQ.

## 3. Materials and Methods

### 3.1. Materials and Reagents

DQ Standard (CAS: 18507-89-6, Purity ≥ 96.8%) was purchased from Dr. Ehrenstorfer DmbH. (Augsburg, Germany). Pyridine (CAS: 110-86-1 analytical grade) was purchased from Shanghai Macklin Biochemical Technology Co., Ltd. (Shanghai, China). Methanol and acetonitrile were HPLC grade and purchased from Merck Inc. (Fairfield, OH, USA). Analytically pure n-hexane, ethyl acetate, acetic anhydride (CAS: 108-24-7), acetic acid (CAS: 64-19-7) and trichloromethane (CAS: 67-66-3) were obtained from Sinopharm Chemical Reagent Co., Ltd. (Shanghai, China). Ultrapure water had a resistivity of 18.25 MΩ * cm (25 C) and used to be grade I water in accordance with the National Standard for Laboratory Water (GB6682-1992).

Organic section nylon needle filters (13 mm × 0.22 μm) were obtained from Shanghai Ampu Experimental Technology Co., Ltd. (Shanghai, China). A chromatographic column (TG-5MS 30.0 m × 0.25 μm × 0.25 mm) was acquired from Thermo Fisher (Waltham, MA, USA). The chromatographic column Strata^TM^-X (60 mg/3 mL, tubes) was acquired from Phenomenex. (Torrance, CA, USA); the extraction column Cleanert PEP (60 mg/3 mL, 50/pkg) was procured from Agilent Technologies. (Beijing, China). The Waters Oasis HLB extraction column (60 mg, 3 mL) was procured from Waters. (Milford, MA, USA).

### 3.2. Standard Stock Solutions and Working Solutions

Prepare a 1.0 mg/mL of DQ standard stock solution in 10 mL of 4% trichloroacetic acid with 10.33 mg of DQ standard (96.8% purity) and store in a refrigerator at −70 °C. The stock solution used to be diluted with chloroform; 100.0 μg/mL, 10.0 μg/mL, and 1.0 μg/mL working solutions of the DQ standard were prepared and saved in a refrigerator at 4 °C.

### 3.3. Preparation of the Samples

#### 3.3.1. Breed of Test Animals and Sample Collection

Twelve 70-day-old chickens (half males and half females) were randomly chosen from Jiangsu Jinghai Poultry Industry Group Co., Ltd. (Nantong, China); they were reared in a single cage with free access to water during the whole feeding period. After one month of feeding, they were slaughtered. The chickens were anesthetized with sodium pentobarbital and sacrificed without delay through guided exsanguination. The chest muscle, leg muscle, liver, and kidney of every poultry were collected, chopped, homogenized, and saved as blank samples in a −34 ℃ fridge for subsequent use.

#### 3.3.2. Sample Extraction

Preliminary sample processing included extraction, purification, and concentration. The homogenized sample (2.0 ± 0.01 g) was precisely weighed in a 50 mL polypropylene centrifuge tube, 5 mL of acetonitrile was added once, and the sample was vortexed for 10 min, ultrasonicated for 10 min, and centrifuged at 5500 rpm for 10 min. The supernatant was transferred into a fresh centrifugal tube. The final residue was repeatedly extracted using 5 mL of acetonitrile and centrifuged at 8000 rpm for 10 min. The two extractive solutions were combined. Ten milliliters of acetonitrile-saturated hexane was added to the extraction solution, vortexed and mixed for 5 min, then centrifuged at 8000 rpm for 5 min, and the fat layer was discarded.

#### 3.3.3. Purification and Concentration

The above samples were dried with nitrogen at 50 °C and then redissolved in 5 mL of 30% acetonitrile solution. The Waters Oasis HLB (60 mg, 3 mL) extraction column was activated and equilibrated with 3 mL methanol and water in turn. The redissolved solution was passed through the chromatographic column at a uniform rate, rinsed with 3 mL of 30% acetonitrile and 3 mL of water, and pumped for 5 min. Finally, the target substance was eluted with 3 mL of acetonitrile. The eluent was collected in a 10 mL centrifuge tube and dried with a nitrogen rate of 50 °C.

### 3.4. Derivatization Reaction

Initially, 1 mL chloroform was added to the purified and concentrated sample for redissolution, and the samples were vortex-blended for 1 min. Then, 300 μL pyridine and 150 μL acetic anhydride were sequentially added and reacted in the dark at room temperature for 3.5 h. After that, the solution was filtered directly via an organic 0.22 μm nylon needle filter, and the filtrate was directly analyzed using GC-MS/MS. Acetic anhydride was used as a derivative reagent and reacted with DQ in the presence of pyridine to generate acetylated DQ in this study, and the reaction is shown in Figure 16.

### 3.5. GC–MS/MS Analysis

This test used to be carried out using a Trace 1300, TSQ 8000 selective MS/MS detector, and a Tripleus RSH autoinjector (Thermo Fisher Scientific Co, Ltd., Waltham, MA, USA) (Table 9). The preliminary temperature was 100 °C; it was held for 1 min, ramped at 30 °C/min to 220 °C, held for 1 min, ramped at 30 °C/min to 290 °C, and held for 13 min.

The data acquisition mode was qualitative with the full scan method and quantitative with the selective reaction monitoring (SRM) method. An electron bombardment ion source (EI) was used in ionization mode with ionization electricity of 70 eV, collision gas of excessive purity argon, and ion source and transmission line temperatures of 280 °C (Table 10).

### 3.6. Quality Parameters

#### 3.6.1. Linearity

The linear regression equation for each matrix used the six standard concentration levels corresponding to the spiked concentrations in blank chicken muscle were LOQ, 25.0, 100.0, 500.0, 1000.0, and 2000.0 µg/kg; the corresponding spiked concentrations in the blank chicken liver and kidney were LOQ, 25.0, 100.0, 1000.0, 2000.0, and 4000.0 µg/kg. Six replicate measurements were made for each concentration, the added concentration of the DQ standard working solution in the three different matrices was used as the abscissa (x), and the peak area of the quantitative ion pair *m*/*z* 231.1 > 230.1* of the DQ was used as the ordinate (y). The determination coefficients and linear regression equations of the standard curves of different matrices were obtained.

#### 3.6.2. LOD and LOQ

Sensitivity in an instrumental analysis is usually expressed with the LOD and LOQ. The LOD is the lowest concentration at which the method can identify the target analyte; the LOQ is the lowest concentration at which the target analyte can be accurately quantified [39]. In this study, blank matrix extracts were prepared with the chicken tissues, and the DQ standard working solution was gradually diluted with the blank matrix extracts and analyzed using the optimized GC-MS/MS method. Six replicates were used for each concentration, and an average signal-to-noise (S/N) value was calculated for each concentration. The DQ concentration corresponding to an S/N ≥ 3 was considered the LOD of the method. The DQ concentration corresponding to an S/N ≥ 10 was considered the LOQ of the method. The recovery rate measured at the LOQ concentration is generally required to be at least 70%, and the RSD should not be higher than 20%.

#### 3.6.3. Matrix Effect

Matrix effects are the effects of non-targeted compounds present in the sample on the detection of the target compound, i.e., the extent to which the matrix interferes with the ability of the analytical method to accurately detect the analyte, which may lead to ion enhancement or inhibition [40]. After establishing the GC/MS/MS method, the slope of the standard curve was used to evaluate the matrix effects. When preparing a sample that matched the standard curve of each matrix, a corresponding solvent standard curve was prepared at the same time. The standard DQ stock solution was gradually diluted with trichloromethane from high to low into standard working solutions of 12.5, 25.0, 500.0, 1000.0, and 2000.0 ng/mL, and derivatization was performed under optimized experimental conditions and analyzed using GC/MS/MS. The peak areas were recorded, and a solvent standard curve was prepared to calculate the matrix effect of DQ in chicken muscle. The standard DQ stock solution was gradually diluted with trichloromethane from high to low into standard working solutions with concentrations of 12.5, 25.0, 100.0, 1000.0, 2000.0, and 4000.0 ng/mL, and the solvent standard curve was used to calculate the matrix effects of the chicken liver and kidney. The matrix effect (ME) calculation formula is [41]: ME (%) = [(Slope _matrix-matched calibration curve_/Slope _solvent standard curve_) − 1] × 100%. If −20% ≤ ME ≤ 20%, a weak matrix effect was observed. Matrix effects present were moderate between −50% ≤ ME ≤ −20% and 20% ≤ ME ≤ 50%. If ME ≤ −50% or ME > 50%, a strong matrix effect was present.

#### 3.6.4. CCα and CCβ

In EU 2002/657/EC [42], the decision limit (CCα) and detection capability (CCβ) are vital indicators for evaluating methods for detecting pesticide residues. CCα is the highest level at which accurate analysis of the target can be performed in a test method and provides the probability that an analyzed test result in a sample does not meet the specified conclusion (false positive). For veterinary drugs with a clearly defined MRL, CCβ refers to the concentration at which the target substance’s MRL concentration can be accurately detected within the sample and the probability that a detected MRL sample can be calculated. For veterinary drugs where the MRL is not specified, CCβ is the lowest concentration of the sample that is able to accurately detect the target and can calculate the probability of negative samples (false negative) that do not meet the specifications. There is a definite MRL for residual DQ in chicken tissues. Twenty blank samples were randomly selected from each matrix, 100 μL of 20.0 μg/mL standard working solution was added to the blank chicken muscle sample, and 100 μL of 40.0 μg/mL standard working solution was added to the chicken liver and kidney samples. The samples were processed in accordance with the pretreatment method, and the standard deviation (SD) was calculated after analysis using the established instrument detection conditions. They are calculated as CCα =MRLs + 1.64 × SD (α = 5%) and CCβ = CCα + 1.64 × SD (β = 5%).

#### 3.6.5. Recovery and Precision

In this study, the accuracy of the approach was investigated with the spiked recoveries of a blank sample and the stability of the method was investigated with the precision of the spiked recoveries of the same batch. The repeatability of the method was verified with the precision of the spiked recoveries of different batches. According to the MRL standards of the US, FDA, and China [12,14], the recoveries of spiked samples for different matrices were calculated at the LOQ, 0.5 MRL, 1.0 MRL, and 2.0 MRL concentration levels chosen inside the linear variation of the matrix standard curve. The reliability of the method was evaluated by calculating the relative standard deviation.

Precisely weighed 2.0 ± 0.02 g of homogenized sample and added DQ standard working solution at four concentration levels of LOQ, 0.5 MRL, 1.0 MRL and 2.0 MRL, and set up six parallel experiments. The samples were processed by methods 3.3 and 3.4, and the obtained derivatives were quantitatively analyzed with the optimized instrument method. The peak areas were used in the matrix standard curve to calculate their corresponding concentrations, and the spiked recovery of every sample was calculated based on the ratio between the actual concentration in the sample and the spiked concentration in the sample. In this study, the RSD was used to evaluate the intraday and interday precision of the samples. The precision was determined from the four added concentrations. The precision test for each added concentration was conducted in three batches in three days. Six replicate samples with the same added concentration were analyzed by the same instrument at different points on the same day, the recovery rate was calculated by the same standard, and the intraday precision was obtained. Interday precision was achieved by analyzing six replicate samples at the same concentration added using the same instrument to create a new matrix standard curve for every day of the week to calculate recovery.

## 4. Conclusions

In this study, we developed a pre-column derivatization-GC-MS/MS method for the determination of DQ residues in chicken tissues. We optimized the sample pretreatment method, including the selection of extraction reagents and derivatization conditions, to improve the reproducibility of analyte determination and extraction efficiency. We developed and optimized the DQ acetylation derivatization reaction., and the derivative was identified as acetylated DQ (C_26_H_37_O_6_N). The recovery rates of the established methods were all above 77.38%. The method was used effectively for quantitative analyses of the residues in real chicken samples, which proved the applicability and feasibility of the method and met the requirements for veterinary drug residue detection. It provides new technical support for the detection of DQ residues in animal foods and meets the needs of different laboratories.

## Figures and Tables

**Figure 1 molecules-28-03875-f001:**
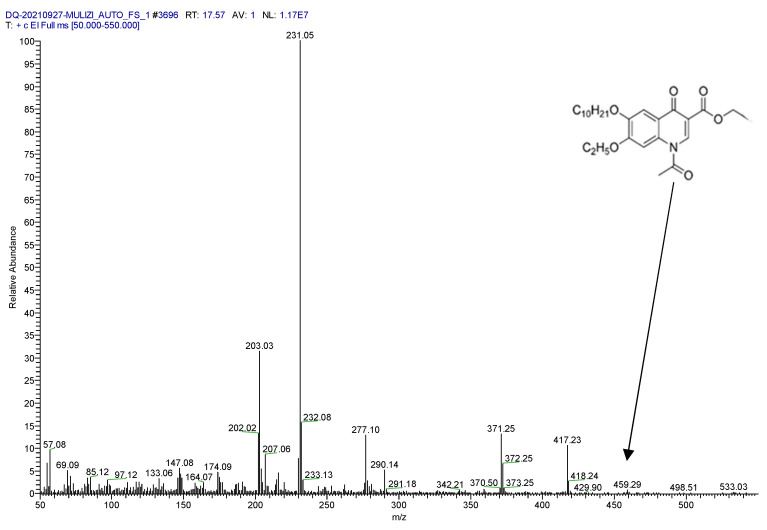
Full-scan mass spectrum of decoquinate derivatives.

**Figure 2 molecules-28-03875-f002:**
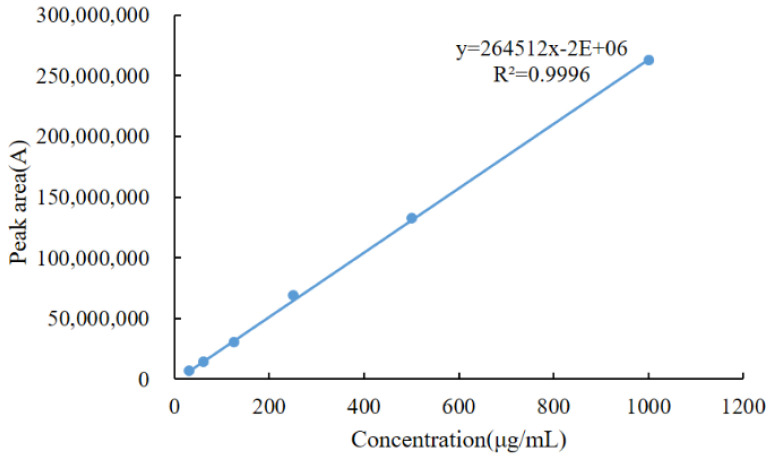
Original drug standard curve.

**Figure 3 molecules-28-03875-f003:**
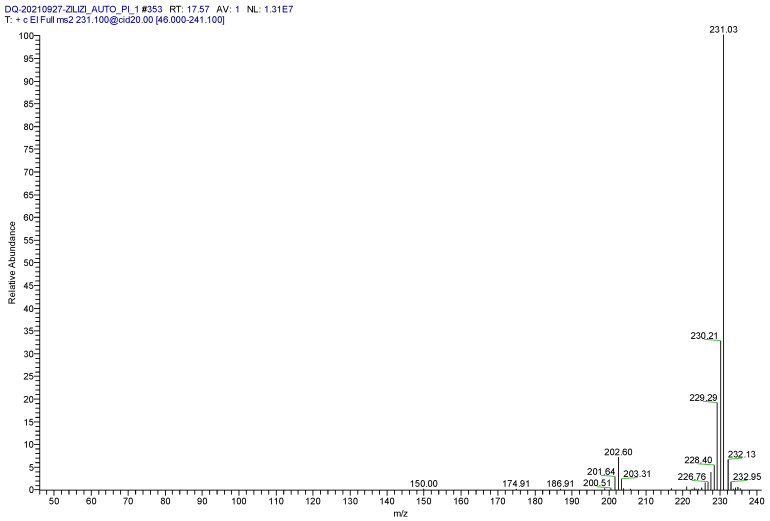
Full scan mass spectrum of the product ion.

**Figure 4 molecules-28-03875-f004:**
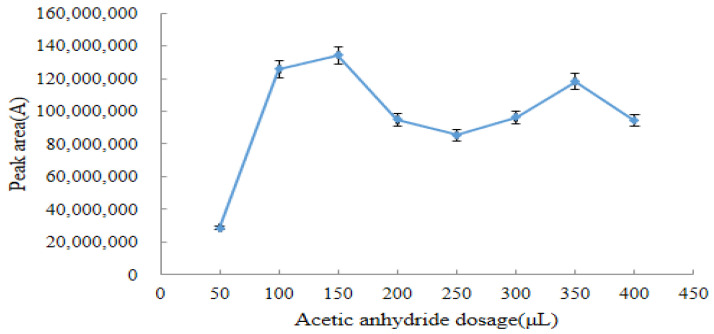
Effect of the acetic anhydride dosage on the response value of the derivatives (*n* = 6).

**Figure 5 molecules-28-03875-f005:**
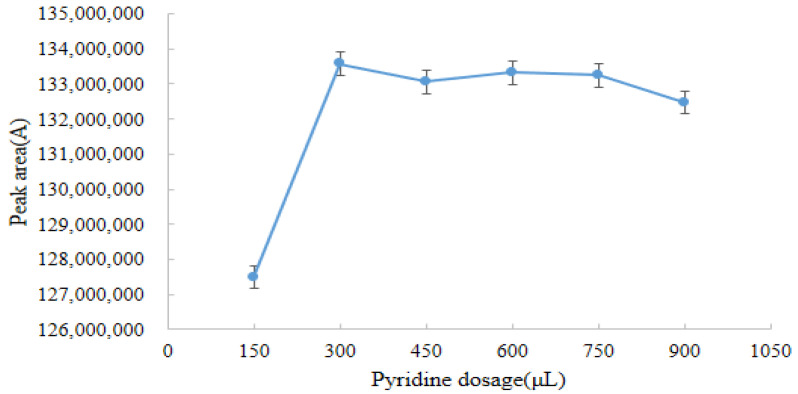
Effect of pyridine dosage on the response value of the derivative (*n* = 6).

**Figure 6 molecules-28-03875-f006:**
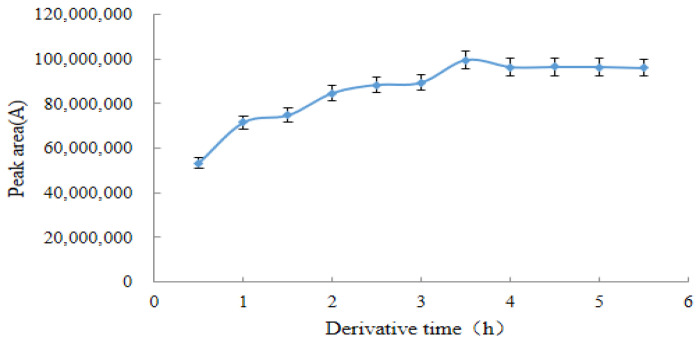
Effect of derivative time on the response value of derivative (*n* = 6).

**Figure 7 molecules-28-03875-f007:**
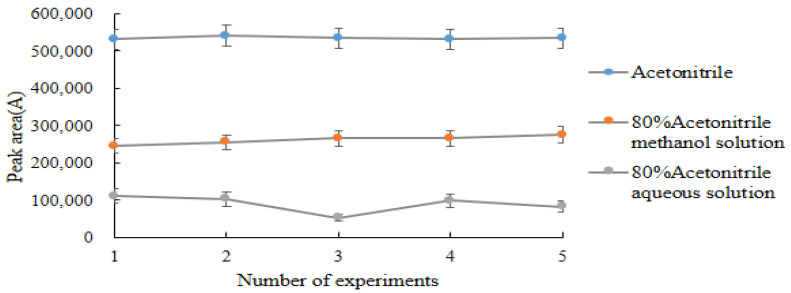
Effects of extraction reagents with different concentrations on the recovery of decoquinate.

**Figure 8 molecules-28-03875-f008:**
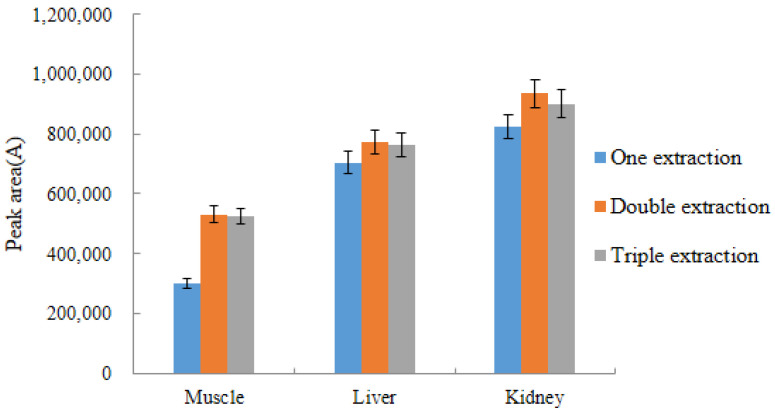
The effect of extraction times on experimental results (*n* = 3).

**Figure 9 molecules-28-03875-f009:**
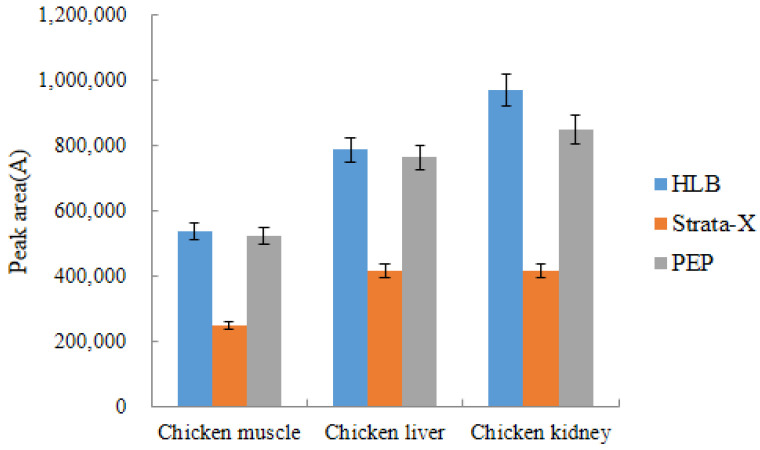
The choice of solid phase extraction column (*n* = 3).

**Figure 10 molecules-28-03875-f010:**
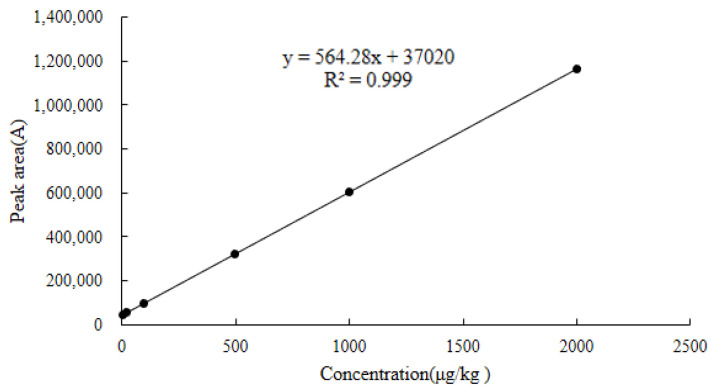
Standard curve for decoquinate in blank chicken muscle.

**Figure 11 molecules-28-03875-f011:**
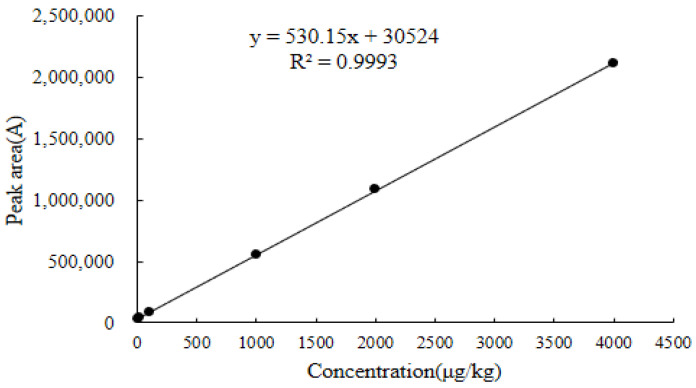
Standard curve for decoquinate in blank chicken liver.

**Figure 12 molecules-28-03875-f012:**
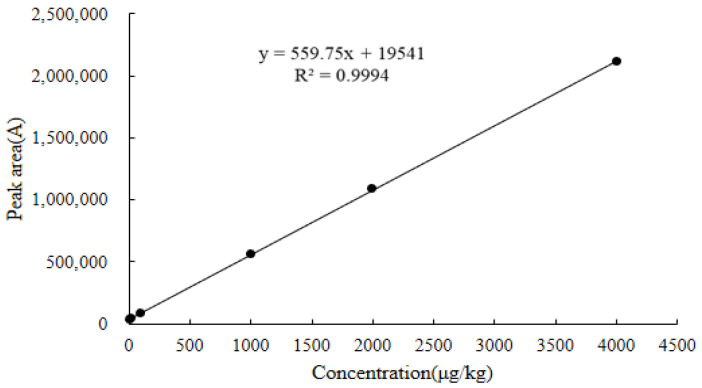
Standard curve for decoquinate in blank chicken kidney.

**Figure 13 molecules-28-03875-f013:**
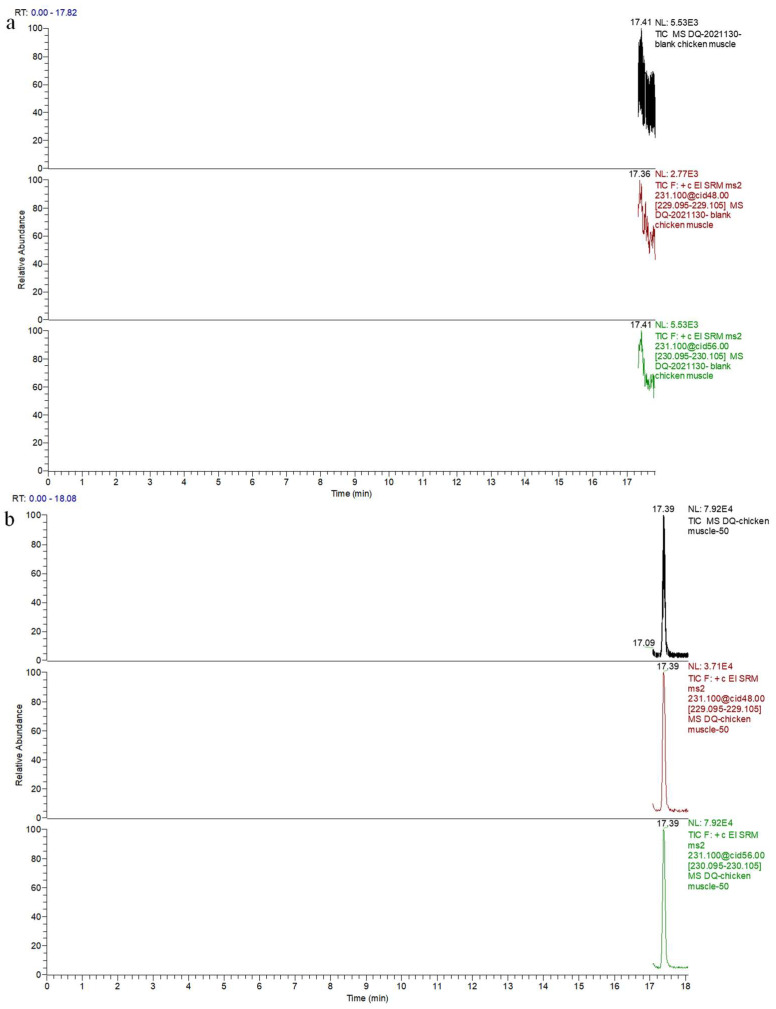
Total ion chromatograms and quantitative and qualitative ion chromatograms of blank chicken muscle and blank muscle spiked with standard. (**a**) Total ion chromatogram and quantitative and qualitative ion chromatograms of blank chicken muscle. (**b**) Total ion chromatogram and quantitative and qualitative ion chromatograms of blank chicken muscle spiked with 100.0 μg/kg decoquinate.

**Figure 14 molecules-28-03875-f014:**
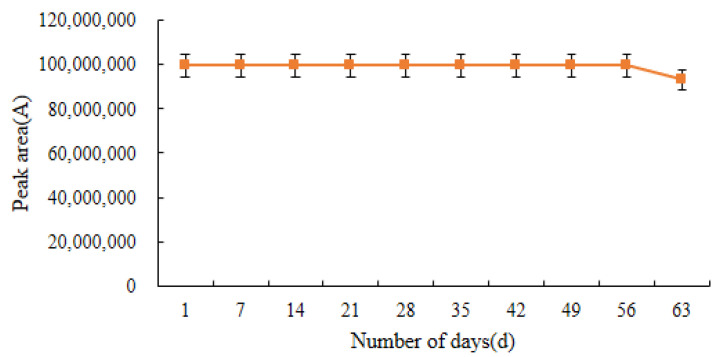
Stability of the decoquinate standard solution stored at −70 °C for 63 days (*n* = 3).

**Figure 15 molecules-28-03875-f015:**
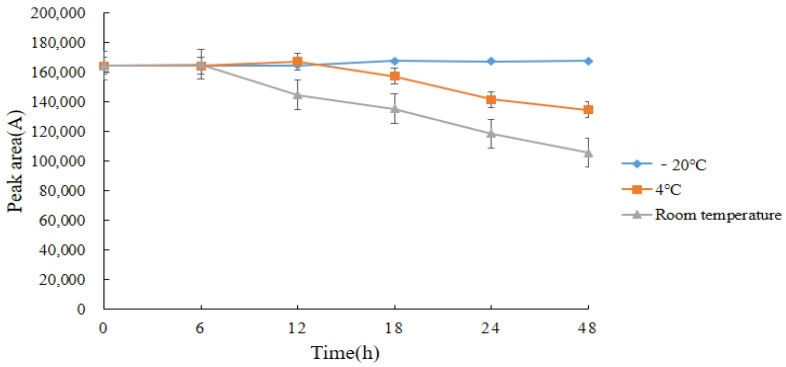
Stability of decoquinate derivatives within 48 h (*n* = 3).

**Figure 16 molecules-28-03875-f016:**
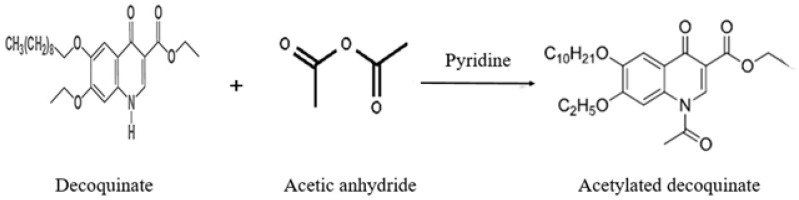
Derivative reaction equation of decoquinate.

**Table 1 molecules-28-03875-t001:** The maximum residue limits (MRLs)values for decoquinate in animal foods established in China, Japan, the USA, and the EU.

Animal Species	Matrix	China (μg/kg)	Japan (μg/kg)	USA (μg/kg)	EU (μg/kg)
Chicken	Muscle	1000	100	1000	-
	Liver	2000	100	2000	-
	Kidney	2000	100	2000	-
	Fat	2000	2000	2000	-
	Other edible viscera	2000	2000	2000	-
Cattle	Muscle	-	1000	1000	-
	Fat	-	2000	2000	-
	Liver	-	2000	2000	-
	Kidney	-	2000	2000	-
	Other edible viscera	-	2000	2000	-
Other terrestrial mammals	Muscle	-	1000	1000 (Goat only)	-
	Liver	-	2000	2000 (Goat only)	-
	Kidney	-	2000	2000 (Goat only)	-
	Other edible viscera		2000	2000 (Goat only)	-

Note: “-” not established or not set.

**Table 2 molecules-28-03875-t002:** Retention time and related MS parameters for the decoquinate derivative target compound.

Target Compound	Molecular Weight	Retention Time (min)	Mass Transitions (*m*/*z*)	Collision Energy (eV)
Acetylated decoquinate	459.57	17.40	231.1 > 229.1 231.1 > 230.1 *	48 56

Note: * Quantificational ion pair.

**Table 3 molecules-28-03875-t003:** Effect of pyridine dosage on the response value of the derivative (*n* = 6).

Matrix	Extraction Method		
	Acetonitrile	Acetonitrile:Ethyl Acetate (1:1, *v*/*v*)	4% Acetic Acid–Acetonitrile Solution
Chicken muscle	87.03 ± 1.02	55.14 ± 2.81	83.26 ± 1.83
Chicken liver	86.02 ± 3.21	41.84 ± 0.63	84.32 ± 1.44
Chicken kidney	85.29 ± 0.76	55.80 ± 1.14	83.67 ± 0.55

**Table 4 molecules-28-03875-t004:** Temperature heating procedure.

Initial Temperature (°C)	Heating Rate (°C/min)	Temperature (°C)	Hold Time (min)
100.0	-	100.0	1.0
100.0	30.0	220.0	1.0
220.0	30.0	290.0	13.0

**Table 5 molecules-28-03875-t005:** Regression equations, determination coefficients, linearity ranges, LODs, LOQs, CCαs, and CCβs for DQ in chicken muscle, liver, and kidney.

Matrix	Linear Regression Equation	Determination Coefficient (R^2^)	Linearity Range (µg/kg)	LOD (µg/kg)	LOQ (µg/kg)	CCα	CCβ
Chicken muscle	y = 564.28x + 37,020	0.9990	4.9–3200	2.2	4.9	1032.0	1034.1
Chicken liver	y = 530.15x + 30,524	0.9993	8.2–4800	4.3	8.2	2042.0	2049.0
Chicken kidney	y = 559.75x + 19,541	0.9994	6.3–4800	3.7	6.3	2047.0	2054.1

**Table 6 molecules-28-03875-t006:** Matrix effects for decoquinate in chicken muscle, liver, and kidney.

Matrix	Solvent Calibration Curve Equation	Matrix Calibration Curve Equation	Matrix Effect
Muscle	y = 668.43x + 53,066, R^2^ = 0.9995	y = 564.28x + 37,020, R^2^ = 0.9990	−15.58
Liver	y = 659.69x + 80,573, R^2^ = 0.9991	y = 530.15x + 30,524, R^2^ = 0.9993	−19.64
Kidney	y = 668.78x + 82,749, R^2^ = 0.9992	y = 559.75x + 19,541, R^2^ = 0.9994	−16.30

**Table 7 molecules-28-03875-t007:** Recovery and precision of decoquinate added to blank chicken muscle, liver, and kidney (*n* = 6).

Matrix	Added Level (μg/kg)	Recovery (%)	RSD (%)	Intraday RSD (%)	Interday RSD (%)
Chicken muscle	4.9	77.38 ± 3.78	4.88	5.09	6.94
	500.0	86.48 ± 3.74	4.32	4.98	5.71
	1000.0 ^α^	85.61 ± 1.28	1.50	2.14	2.27
	2000.0	88.33 ± 1.45	1.64	3.02	3.44
Chicken liver	8.2	78.33 ± 2.14	2.73	5.56	6.62
	1000.0	84.78 ± 2.20	2.59	3.41	5.07
	2000.0 ^α^	85.15 ± 1.17	1.37	2.26	3.47
	4000.0	89.65 ± 3.63	4.05	4.91	5.30
Chicken kidney	6.3	79.15 ± 3.43	4.33	5.74	8.06
	1000.0	87.95 ± 1.01	1.15	2.10	2.47
	2000.0 ^α^	88.59 ± 1.03	1.16	1.63	3.20
	4000.0	89.10 ± 2.71	3.04	4.09	4.53

Note: “*α*” Maximum residue limits.

**Table 8 molecules-28-03875-t008:** Comparison of detection methods for performance parameters of decoquinate in different animal foods.

Matrix	Analytical Method	Chromatographic Conditions	LODs (μg/kg)	LOQs (μg/kg)	Recovery Rate (%)
Chicken liver [33]	HPLC-UV	Agilent Eclipse XDB-C_18_ (4.6 × 250 mm, 5 μm) Mobile phase: Acetonitrile-ethyl acetate (1:1, *V*/*V*)	100	200	72.9~96.8
DQ capsule [31]	HPLC-UV	Agilent ZORBAX 80A Extend-C_18_ (25 × 4.6 mm, 5 μm) Mobile phase: acetonitrile-0.1% formic acid aqueous solution (75:25, *V*/*V*)	-	50	≥98
Chicken muscle [36]	LC-MS/MS	XTerra MS (C_18_: 2.1 × 100.0 mm, 3.5 μm) Mobile phase: A: 0.1% formic acid aqueous solution; B: acetonitrile	1.0	2.5	85.3~104.9
Milk [24]	HPLC-MS/MS	Agilent Zorbax Eclipse Plus C18 RRHD (50.0 × 2.1 mm, 1.8 µm) Mobile phase: A: 0.01 moL ammonium formate buffer (pH 4.0); B: acetonitrile	0.78	5.0	≥98.3
Eggs [23]	LC-MS/MS	Zorbax 80 SB-C_18_ (2.1 × 150 mm, 5 μm) Mobile phase: A: methyl alcohol; B: acetonitrile; C: 0.01 mol/L ammonium formate buffer (pH 4.0)	0.09	0.36	80.0~120.0
Chicken muscle [37]	LC-MS/MS	Agilent Poroshell 120 EC-C18 (2.1 × 100 mm,2.7 μm) Mobile phase: A: methyl alcohol; B: 1% acetic acid and 5 mmoL ammonia buffer	0.008	0.027	74~112
Fodder [8]	HPLC-FLD	C_18_ (250 × 4.6 mm, 5 μm) Mobile phase: 0.005 mol/L calcium chloride solution- methyl alcohol (15:85, *V*/*V*)	-	500	82.5~99.0
Bovine liver [34]	UHPLC-MS/MS	Acquity BEH C18 (50.0 × 2.1 mm i.d., 1.7 μm) Mobile phase: A: 0.005 moL/L aqueous ammonium acetate solution (containing 0.05% formic acid); B: acetonitrile	5	10	78~102
Fodder [38]	SERS	-	381		-
Chicken muscle [32]	HPLC-MS	Acquity BEH C18 (50.0 × 2.1 mm, 1.7 μm) Mobile phases: A: 0.005 mol/L ammonium acetate solution (contained 0.05% formic acid); B: methyl alcohol	5	50	≥84.8
Chicken muscle, liver, and kidney (this study)	GC-MS/MS	TG-5MS AMINE (30.0 m × 0.25 mm × 0.25 μm) Mobile phase: high purity helium (99.999%, 60 psi)	2.2~4.3	4.9~8.2	77.38~89.65

Note: “-” not reported.

**Table 9 molecules-28-03875-t009:** Gas chromatography conditions.

Instrument Content	Experimental Conditions
Chromatographic column	TG-5MS (30.0 m × 0.25 μm × 0.25 mm i.d.)
Carrier gas	Helium (99.999%, 60 psi)
Carrier gas mode	Constant current mode
Carrier gas column velocity	1.0 mL/min
Carrier gas saves time and flow	Time: 2 min, flow rate: 20.0 mL/min
Injection port temperature	280 °C
Shunt mode	Undivided injection
Shunt flow	50.0 mL/min
Non shunting time	1.0 min
Injection volume	1.0 μL
Temperature programmed	Initial temperature 100 °C, maximum temperature 290 °C

**Table 10 molecules-28-03875-t010:** Mass spectrometry conditions.

Instrument Content	Experimental Conditions
Ionization mode	Electron bombardment ion source (EI)
Electron beam energy	70 eV
Collision gas	High purity argon (>99.999%, 40 psi)
Ion source temperature	280 °C
Transmission line temperature	280 °C
Solvent delay	3.0 min
Acquisition data mode	SCAN mode is qualitative Auto SRM mode is quantitative

## Data Availability

Data will be made available on request.

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
