# Peer review of "Qualitative and Quantitative Determination of Decoquinate in Chicken Tissues by Gas Chromatography Tandem Mass Spectrometry"

_molecules, 2023, doi:10.3390/molecules28093875_

Round 1

Reviewer 1 Report

This study described the qualitative and quantitative determination of decoquinate in chicken tissues by gas chromatography tandem mass spectrometry.   The study was validated and applied for real sample analysis form super market. Authors claim novelty as this is first method for analysis of target analyte by GC-MS/MS. The manuscript fits within the scope of the journal and is also interesting. However, the following need to be addressed properly for further improvement of the manuscript. They are: 

There are too many abbreviations in the Abstract which should need to avoid. Kindly reduce or remove them.

Authors tried to present the structure of the article at the end of the introduction. However, it is too much and not well structured. Kindly improve on that

The introduction section could be improved. Author should refer to articles from reputable journals to enrich and enhance the introduction section of the manuscript specially comparative advancement of this method in term of LOD/LOQ, speed and simplicity,

There are several formatting and syntex issues in the manuscript. Kindly work on them. Example: line 175

The paper needs more effort in displaying the methodology as the application is not so innovative. It lacks information of different indices used in this research. This section should be re-organized, more specific and systematized with references to support it.

Why the author performs both LLE and SPE together for the sample analysis. How the acetonitrile can work as extractant for sample extraction?

Discussion of results is weak. It is suggested to compare the results of the present study with some similar studies.

It is suggested to re-organize the conclusion section much better. The conclusion is too much and lacks some basic components. Kindly summarize. It should be re-written in a well-structured manner. It should cover a summary of the problem(s), objectives methodology, findings, and recommendation(s).

Author Response

We greatly appreciate the efficient, professional and rapid processing of our contribution (Manuscript Number: molecules-2336585) by the editor, and we also appreciate the meticulous work of the assigned editor in reviewing our previous manuscript.

We modified the manuscript with point-by-point revisions based on the suggestions and comments of the reviewers, and all revisions in the current version are highlighted in red font. We hope that all revisions are valuable and will improve the level of the manuscript to meet the standards of the reviewers and the requirements of the journal.

The detailed responses are as follows:

Response to Reviewer Comments

Point 1: There are too many abbreviations in the Abstract which should need to avoid. Kindly reduce or remove them.

Response 1: Thank you for your professional advice. Based on your comments, the abstract has been rewritten, please refer to lines 22-23 in the new manuscript.

Point 2: Authors tried to present the structure of the article at the end of the introduction. However, it is too much and not well structured. Kindly improve on that

The introduction section could be improved. Author should refer to articles from reputable journals to enrich and enhance the introduction section of the manuscript specially comparative advancement of this method in term of LOD/LOQ, speed and simplicity.

Response 2: Thank you for your professional advice. In response to your suggestion, I have improved the introduction. Please refer to lines 52-68 in the new manuscript.

Point 3: There are several formatting and syntex issues in the manuscript. Kindly work on them. Example: line 175

Response 3: Thank you for your comment. I have revised the format and syntax of this section; please refer to lines 458-460 in the new manuscript.

Point 4: The paper needs more effort in displaying the methodology as the application is not so innovative. It lacks information of different indices used in this research. This section should be re-organized, more specific and systematized with references to support it.

Response 4: Thank you for your professional advice. I have revised the methodology and added relevant references. Please refer to lines 425-428, 435-438, 444-445, and 447-450 in the new manuscript.

Point 5: Why the author performs both LLE and SPE together for the sample analysis?

Response 5: Thank you for your question. The extraction reagents commonly used in LLE are organic solutions, which easily extract interfering components from the sample and affect the accuracy of the experimental results. SPE mainly uses solid adsorbents to adsorb the target substances and separate impurities and target substances in the samples to achieve purification. Therefore, the combination of LLE and SPE enabled further separation of the interfering components from the target prior to instrument analysis, reduced the matrix effects, protected the column, and extended its useful life.

Point 6: How the acetonitrile can work as extractant for sample extraction?

Response 6: Thank you for your question. According to the physicochemical properties of the drugs, we found after consulting many studies that the commonly used extraction reagents were mainly acetonitrile, trichloromethane, and ethyl acetate. The effects of acetonitrile, acetonitrile:ethyl acetate and 4% acetonitrile acetate solution in extracting DQ from chicken tissues were compared. The results showed that the recovery seen when acetonitrile was used as the extraction agent was higher than those of the other two solvents. Therefore, acetonitrile was selected as the extraction reagent in this experiment.

Point 7: Discussion of results is weak. It is suggested to compare the results of the present study with some similar studies.

Response 7: Thank you for your professional advice. I have modified the discussion section based on your comments; please refer to lines 318-326 and 334-338 in the new manuscript.

Point 8: It is suggested to re-organize the conclusion section much better. The conclusion is too much and lacks some basic components. Kindly summarize. It should be re-written in a well-structured manner. It should cover a summary of the problem(s), objectives methodology, findings, and recommendation(s).

Response 8: Thank you for your professional advice. I have rewritten the conclusions section; please refer to lines 511-521 in the new manuscript.

Reviewer 2 Report

The present manuscript makes a fully description of a new method to determine decoquinate in chicken tissues by GC-MS/MS. However, it lacks in highlighting the novelty of the presented work. There are currently some published methods that can be used to determine several coccidiostats at once, by LC-MS/MS without the derivatization step. Despite the factor that authors claim that using GC can be less expensive than HPLC, the possibility of analyzing more compounds at once is a huge advantage. Also, in table 10 where some methods are being compared, it can be seen than other publications presented methods with lower LOD and LOQ. So, the authors should provide their ideas on why this is a study worth to be published and to be used by others.

The writing is clear and some remarks can be considered to improve the final document:

In table 1, where the maximum residue limits are presented, the symbol “-“ should be designated as “not established or not set” and not “not detected”. In fact, at least for EU legislation, there is presented as “no MRL required”.

In the session Materials and Reagents, please try to avoid the use of the same word “purchased” repeatedly (8 times).

Line 87 and 89: The normal designation is stock solutions (inventory solution is not the appropriated term).

Line 100: Is there any knowledge concerning the stability of the compound in the sample? How long can it be stored before analysis?

Line 146: What is the meaning of the following sentence “The concentration of DQ in blank chicken muscle was the limit of quantitation (LOQ)”? Please clarify.

Line 181: Since 2021 that the EU 2002/657/EC is no longer in place. Authors should analyse the new Commission Implementing Regulation (CIR) 808/2021 since there are some modifications in the CCα and CCβ evaluation.

Overall, in my opinion, there are some tables that can be linked. For instance, the results of the method validation could be in one single table.

Author Response

We greatly appreciate the efficient, professional and rapid processing of our contribution (Manuscript Number: molecules-2336585) by the editor, and we also appreciate the meticulous work of the assigned editor in reviewing our previous manuscript.

We modified the manuscript with point-by-point revisions based on the suggestions and comments of the reviewers, and all revisions in the current version are highlighted in red font. We hope that all revisions are valuable and will improve the level of the manuscript to meet the standards of the reviewers and the requirements of the journal.

The detailed responses are as follows:

Response to Reviewer Comments

Point 1: The present manuscript makes a fully description of a new method to determine decoquinate in chicken tissues by GC-MS/MS. However, it lacks in highlighting the novelty of the presented work. There are currently some published methods that can be used to determine several coccidiostats at once, by LC-MS/MS without the derivatization step. Despite the factor that authors claim that using GC can be less expensive than HPLC, the possibility of analyzing more compounds at once is a huge advantage. Also, in table 10 where some methods are being compared, it can be seen than other publications presented methods with lower LOD and LOQ. So, the authors should provide their ideas on why this is a study worth to be published and to be used by others.

Response 1: Thank you for your professional advice. In this study, we proposed and optimized for the first time an acetylation derivatization reaction for DQ, identified the derivative as acetylated DQ, and proposed a new qualitative and quantitative scheme to improve the determination and monitoring method of DQ trace levels, which provides technical support to meet the needs of different laboratories or testing units.

Point 2: In table 1, where the maximum residue limits are presented, the symbol “-“ should be designated as “not established or not set” and not “not detected”. In fact, at least for EU legislation, there is presented as “no MRL required”.

Response 2: Thank you for your professional advice. I have modified this symbol "-" in the new manuscript, please refer to Table 1 in the new manuscript.

Point 3: In the session Materials and Reagents, please try to avoid the use of the same word “purchased” repeatedly (8 times).

Response 3: Thank you for your professional advice. In response to your comment, the corresponding content has been revised in the manuscript. Please refer to lines 356-365 in the new manuscript.

Point 4: Line 87 and 89: The normal designation is stock solutions (inventory solution is not the appropriated term).

Response 4: Thank you for your professional advice. I have modified the term "inventory solution" in the manuscript. Please refer to line 367 of the new manuscript.

Point 5: Line 100: Is there any knowledge concerning the stability of the compound in the sample? How long can it be stored before analysis?

Response 5: Thank you for your question. In this study, Haiyang yellow chickens were fed full-priced feed without any drugs during the feeding period, and the chicken tissue samples were stored in a -34 °C refrigerator as blank test samples and exhibited stable storage for 2 months.

Point 6: Line 146: What is the meaning of the following sentence “The concentration of DQ in blank chicken muscle was the limit of quantitation (LOQ)”? Please clarify.

Response 6: Thank you for your question. We have reworded the sentence in the new manuscript; please refer to lines 425-428.

Point 7: Line 181: Since 2021 that the EU 2002/657/EC is no longer in place. Authors should analyse the new Commission Implementing Regulation (CIR) 808/2021 since there are some modifications in the CCα and CCβ evaluation.

Response 7: Thank you for your professional advice. Prior to preparing this manuscript, I reviewed and studied (CIR) 808/2021. However, this study completed verification of the parameters before 2021. Meanwhile, I noted in the reference of (CIR) 808/2021 that“until 10 June 2026, the requirements laid down in points 2 and 3 of Annex I to Decision 2002/657/EC shall continue to apply to methods, which have been validated before the date of entry into force of this Regulation”. Therefore, we referred to the EU 2002/657/EC.

Round 2

Reviewer 1 Report

Authors addressed most of the issue properly but some concern still existing in the manuscript.

1. Abbreviation (SPE and LLE) still existing in the abstract section even though mentioned in the response that it was corrected.

2. I still not convinced that acetonitrile can be used as extractant in LLE method. If some reference existing then it should be cited. LLE method is based on partition coeffcient principle. Kindly clarify it
